# Genetically Encoded ATP Biosensors for Direct Monitoring of Cellular ATP Dynamics

**DOI:** 10.3390/cells11121920

**Published:** 2022-06-14

**Authors:** Donnell White, Qinglin Yang

**Affiliations:** 1Cardiovascular Center of Excellence, Louisiana State University Health Sciences Center, New Orleans, LA 70112, USA; dwhi20@lsuhsc.edu; 2Department of Pharmacology and Experimental Therapeutics, School of Graduate Studies, Louisiana State University Health Sciences Center, New Orleans, LA 70112, USA; 3School of Medicine, Louisiana State University Health Sciences Center, New Orleans, LA 70112, USA

**Keywords:** ATP, ATP dynamics, cellular bioenergetics, energy metabolism, mitochondria, glycolysis, real-time ATP monitor, genetically encoded fluorescent biosensors, spatiotemporal

## Abstract

Adenosine 5′-triphosphate, or ATP, is the primary molecule for storing and transferring energy in cells. ATP is mainly produced via oxidative phosphorylation in mitochondria, and to a lesser extent, via glycolysis in the cytosol. In general, cytosolic glycolysis is the primary ATP producer in proliferative cells or cells subjected to hypoxia. On the other hand, mitochondria produce over 90% of cellular ATP in differentiated cells under normoxic conditions. Under pathological conditions, ATP demand rises to meet the needs of biosynthesis for cellular repair, signaling transduction for stress responses, and biochemical processes. These changes affect how mitochondria and cytosolic glycolysis function and communicate. Mitochondria undergo remodeling to adapt to the imbalanced demand and supply of ATP. Otherwise, a severe ATP deficit will impair cellular function and eventually cause cell death. It is suggested that ATP from different cellular compartments can dynamically communicate and coordinate to adapt to the needs in each cellular compartment. Thus, a better understanding of ATP dynamics is crucial to revealing the differences in cellular metabolic processes across various cell types and conditions. This requires innovative methodologies to record real-time spatiotemporal ATP changes in subcellular regions of living cells. Over the recent decades, numerous methods have been developed and utilized to accomplish this task. However, this is not an easy feat. This review evaluates innovative genetically encoded biosensors available for visualizing ATP in living cells, their potential use in the setting of human disease, and identifies where we could improve and expand our abilities.

## 1. Introduction

Adenosine 5′-triphosphate (ATP) is perhaps the most utilized and widely applicable molecule of all living organisms [1]. ATP is formed in oxidative reactions within the mitochondria and through glycolysis. Mitochondria are responsible for housing numerous metabolic pathways, such as the citric acid cycle and fatty acid oxidation [2,3]. Oxidative phosphorylation, or OXPHOS, occurs in the mitochondrial respiratory chain composed of the electron transport chain and ATP synthase. The electron transport chain comprises four complexes that oxidize NADH, H^+^, and FADH_2_, ultimately resulting in electrons being transferred to O_2_, thus making this process aerobic. This sequential progression of oxidation of donor molecules results in a proton gradient, forming an electrochemical ion gradient across the inner mitochondrial membrane (IMM). The outer side of the IMM is positively charged (the p-side), and the inner side is negatively charged (the n-side) [4]. This gradient provides the potential needed for the leading producer of ATP, F_1_F_0_ ATP synthase, to generate ATP from the phosphorylation of ADP and inorganic phosphate [1,5,6]. This ATP is transferred to different cellular compartments for cellular functionalities.

Along with being the primary energy currency of the cell, ATP also has numerous other functions, ranging from neuronal membrane potential [7], organelle transport [8], muscle contraction [4], ion exchanges [6,9,10], signaling ligand for G-protein coupled receptors [11], and translocation of materials between the nucleus and cytosol [12]. ATP also plays critical roles in inflammatory pathways, and cyclic adenosine monophosphate (cAMP) signaling that plays a role in cancer growth [13,14]. Additionally, ATP is a purinergic neurotransmitter, acting on both pre- and post-junctional ionotropic purinergic receptors. Specifically, ATP is used in most peripheral and central neurons to assist with neurotransmission and neuromodulation of autonomic and sensory ganglia in the brain and spinal cord [15]. ATP is also shown to assist with enteric neuron communication [16]. Information of real-time dynamic ATP changes should further define the role of ATP in regulating a variety of functions.

Numerous methodologies have been employed for in vitro, ex vivo and in vivo ATP detection and measurement. The in vivo and ex vivo measurement of ATP is involved in the use of magnetic resonance spectroscopy (MRS) and can only quantify total ATP signals from whole tissues [17]. Most of the in vitro ATP detection techniques based on chromatography and chemiluminescence require homogenized tissues or cells. The luciferase-based bioluminescence techniques were developed to measure ATP contents in tissue and cell lysates, and were eventually extended to living cells [18]. Given the crucial role of energy metabolism, numerous methodologies have been developed and extensively used to gain a better understanding of the bioenergetic changes occurring within living cells [19,20]. Oxography is a technique that allows for high-resolution respirometric measurement of bioenergetics in permeabilized cells and isolated mitochondria in response to substrates, electron transport chain (ETC) complex inhibitors, and other metabolic effectors [21,22]. Oxygraphy measures oxygen (O_2_) as a gas dissolved in an aqueous solution, monitoring oxygen consumption and flux over time within the sample. Oxygraph also provides the capability to provide simultaneous real-time monitoring of reactive oxygen species (ROS) production [23]. While this method provides in-depth insights into bioenergetic processes, it remains challenging to determine the real-time ATP change even in the mitochondria. Seahorse Bioanalyzers are another metabolic flux technology used to analyze cellular bioenergetics, which has been extensively used in recent years for cellular metabolism studies. The Seahorse Bioanalyzer allows for high-throughput simultaneous kinetic measurements of respiration and glycolysis, as well as indirect estimates of the ATP production [24,25,26]. Although data collected through these methods provide insight into the bioenergetics profiles of isolated tissues or cells, they provide very limited visual indications of ATP usage within specific compartments. Unfortunately, they cannot decipher whether some compartments use or deplete ATP more than others [27]. Direct ATP monitoring is apparently the preferable approach to understanding cellular ATP dynamics in various physiologic and pathologic conditions. In the following sections, we will discuss why detailed information regarding spatiotemporal cellular ATP dynamics is crucial in understanding cellular function under physiological and pathological conditions. We will further delineate how innovative genetically encoded ATP biosensors will fulfill this goal.

## 2. ATP Dynamics in Healthy and Diseased Cells

Due to functionality differences, different cell types throughout the body and across different organisms at different stages exert different ATP demands, hence varying ATP production and dynamics. It is crucial to have a reliable method to evaluate the unique dynamics of a given cell type. A clear picture of ATP spatiotemporal dynamic movement is vital for understanding the biochemical basics of diseases affecting various cell types, providing insights for more targeted treatments of these diseases. First, it is essential to understand the basics of ATP movement in and out of the mitochondria. Mostly, mitochondrial carrier proteins are responsible for the transport of ATP across mitochondrial membranes. The major transporters in humans are the ADP/ATP carriers (AAC or Adenine Nucleotide Translocator, ANT) (Figure 1). AAC resides within the inner mitochondrial membrane, exporting ATP and importing ADP [28,29,30] (Figure 1). The voltage-dependent anion channel (VDAC) on the outer membrane of mitochondria further transports ATP out of the cell into the cytoplasm, where it can be utilized by the cell for its numerous functions (Figure 1). When the cellular metabolism is influenced by external factors, cellular modifications and adaptations must take place to ensure the movement and transport of ATP remains consistent. However, most of the current views regarding the dynamic ATP trafficking between cellular compartments are not based on direct visualization due to the lack of feasible methodologies.

Transient hypoxia, or low O_2_ for a brief period, stimulates tightly regulated survival mechanisms within the cell, explicitly initiating the AMP-activated protein kinase (AMPK) pathway [31]. Hypoxia increases glycolytic ATP production through the increased enzymatic activity of phosphofructokinase-1 and pyruvate kinase. This uptick in glycolysis is a compensation response, as oxygen shortage stimulates the cells to consume glucose faster to produce ATP through anaerobic glycolysis. This compensation is also called the Pasteur Effect [32,33]. Under these conditions, mitochondria are not fueled with substrates (Acetyl CoA and O_2_), further inducing structural and functional changes [34,35]. However, this short-term survival response to transient hypoxia is only briefly sustainable. When oxygen concentrations are low for extended periods, cells stimulate the activation of hypoxia-inducible factors (HIFs). HIFs are heterodimeric transcription factors composed of α and β subunits [35,36,37,38]. HIFs bind to hypoxia-responsive elements (HREs), thus activating gene transcription (such as adenylate kinase 3, GLUT1, GLUT3, HK1, HK2, LDH A, and PFK) to equip the cell with long-term adaptations to survive in their hypoxic environment [35,37,38,39,40]. This adaptation results in a shift in ATP production and variations in concentrations of ATP across organelles [41]. Different types of cells display varying levels of ATP dynamics, indicating various basal levels of mitochondria oxygen consumption [42]. Thus, understanding the basics of ATP movement and dynamics across various cell types is vital in clarifying the underlying specific pathological conditions in various tissues and cells [4]. When faced with decreased ATP availability, metabolic reprogramming can lead to pathologies and potential death [20,42,43,44,45]. Specifically, it is established that failure of ATP homeostasis can lead to diseases such as cardiovascular disease, neurodegenerative disorders, and cancer [7,46,47,48]. Real-time evaluation of ATP dynamics within cells using ATP biosensors can help reveal mechanisms underlying these diseases that previously would not have been possible.

The heart obtains 95% of its energy from ATP to support the contraction–relaxation cycle within the myocardium [49,50,51]. Approximately 70% to 90% of ATP in adult cardiomyocytes is produced by the oxidation of fatty acids, with the remaining 10% to 30% coming from glucose oxidation, lactates, and small amounts of ketone bodies and amino acids [49,52,53,54,55]. Cardiac tissue has a high rate of ATP hydrolysis and if the synthesis of ATP were inhibited, ATP reserves would only sustain the contraction–relaxation cycle for a few seconds [51,52,56]. A well-functioning heart requires maintaining a steady balance of ATP supply and demand. Conversely, a failing heart requires significantly more ATP while in a state of increasingly limited O_2_, leading to metabolic rearrangement and exacerbating pathologic remodeling and contractile dysfunction. Intense deviations and depletion of ATP levels result in the inability to maintain adequate contractility over time [51]. ATP generated from glycolysis contributes as little as 5% of the total ATP consumed in an average adult heart [57]. The heart shifts to use glucose as the main substrate for the production of ATP, though glycolysis alone is ineffective in providing the energy needed to compensate for the loss of mitochondrial ATP production [58]. However, the ATP content is primarily maintained until the end stage of heart failure [27,59,60]. One reason for this might be that myocardial energy is maintained through adaptive mechanisms that are poorly understood [61,62]. Recent advances in the innovation and development of ATP biosensors may provide a feasible approach to solving some of the puzzles. With the ability to monitor the ATP movement and flux over time in living cardiomyocytes, ATP can be visualized in various sub-compartments when subjected to various stress treatments that closely mimic common pathological states in the heart. By having the capability to track the movement and dynamics of ATP in these conditions, it can be evaluated how different aspects of energy metabolism adapt to the stresses and which cellular compartments are contributing to the preservation of cardiac function. The insight revealed through carefully designed experiments can provide unprecedented mechanistic insights. A study conducted by Luptak represents one of the first instances in which ATP synthesis is shown to be decreased in metabolic heart disease [63]. This experiment was accomplished via 31p nuclear magnetic resonance and thin-layer chromatography. However, one major limitation in this study is that the ATP-free energy calculated represents the average value across the heart. Thus, the values do not reflect cardiomyocyte variations, especially those in subcellular compartments. The technological limitations in this well-designed study provide an example of a situation in which compartmental-specific ATP biosensors could evaluate differences in ATP levels across cardiomyocytes. Thus, genetically encoded biosensors provide the opportunity to answer the fundamental and underlying pathological questions that will lead to a better understanding and treatment of these diseases.

Through the use of in vitro experimental models of hypoxia–reoxygenation, it is possible to investigate the cardioprotective methods in which the cardiomyocytes regulate and moderate their own hypoxia-induced autophagy in these conditions [64]. As hypoxia is well known to induce autophagy, mitochondrial pathways are a key factor responsible for this pathophysiology [65]. Additionally, hypoxia-induced cell injuries can be further aggravated by the reoxygenation [64,65,66,67], begging for additional questions to be asked and answered. When evaluating cardiomyocyte metabolic function in hypoxic conditions, there are a few obstacles that must be faced in doing so. Mainly, there are few reliable technologies by which to monitor ATP usage in hypoxic situations. For example, luciferin–luciferase technology cannot be used in these conditions, as luciferin requires oxygen to undergo its enzymatic reaction, thus making the reaction aerobic [68]. Genetically encoded ATP biosensors can help to solve this issue, as this biosensor can be packaged into adenovirus and efficiently be transduced into the cardiomyocytes [69]. Moreover, these biosensors bind to ATP directly, fluorescing due to changes in protein conformation. This minimizes errors associated with other enzymatic reactions that may be occurring, or the biosensor interacting with other metabolites. A more in-depth discussion regarding how biosensors work will be discussed later.

Furthermore, cardiac hypertrophy is a major risk for heart failure [70,71]. Based on mostly in vivo and ex vivo studies on animal models with pathological cardiac hypertrophy or heart samples from patients of heart failure, mitochondrial dysfunction and impaired energy metabolism are among the main changes in hypertrophic and failing hearts [56,63,72,73,74]. With the emerging of innovative technology such as the Seahorse Bioanalyzer, cellular bioenergetics can now be directly evaluated in cultured cardiomyocytes [75,76]. On the other hand, disease models of cardiomyocytes, such as cardiomyocytes from animals, tumor-derived cell lines and human-induced pluripotent stem cell-derived cardiomyocytes (hiPSC), have been extensively used in cardiac research. Various hypertrophic stimuli are applied, including mechanical stretch [77], Angiotensin II (Ang II) [78,79], phenylephrine (PE) [80,81,82], and endothelin 1 [83,84]. Real-time investigations of ATP production and dynamics in addition to examining metabolic flux on clinically relevant cellular disease models become plausible using these techniques. The utility of genetically encoded ATP biosensors should provide a useful approach to determine how energy metabolism perturbs during the pathological development when cardiomyocytes are subjected to pathological stimuli. Direct evaluation of innovative therapies on energy metabolism on cardiomyocytes will provide unprecedented opportunities to understanding the pathological mechanisms and potential effective treatments.

Another area in which these biosensors could provide in-depth insights is utilizing them in cardiomyocytes to evaluate metabolic function in the setting of comorbidities. As obesity and type II diabetes are becoming increasingly more prevalent across the globe, particularly in the United States, it would be a disservice to neglect a discussion about their comorbidity associated with cardiovascular disease [85,86]. In diabetic patients, hyperglycemia and hyperlipidemia induce both morphologic and physiological changes to cardiac tissue, thus leading to diabetic cardiomyopathy [87]. It has also been shown that an increased number of circulating fatty acids induce cardiomyocyte apoptosis, while glucose has not been shown to cause direct activation of the apoptotic pathway. However, both nutrients in excessive quantities inhibit proper formation and maintenance of contractile structures, which is pertinent for the contraction–relaxation cycle [88]. Even more so, diabetes also causes calcium dysregulation in cardiomyocytes, leading to detrimental effects, inhibiting the cardiomyocytes’ contractile ability [89]. This information emphasizes that cardiomyocytes, under additional stressors from comorbidities such as diabetes, walk a stringent metabolic tightrope in the production and depletion of ATP. Incorporating a technique to evaluate the metabolic changes overtime after stimulating diabetes in a cardiomyocytes may lead to insight into the aggressive correlation seen between cardiovascular disease and diabetes. Obesity, typically hand-in-hand with type II diabetes, is another common factor associated with cardiovascular disease [85,90,91]. Obesity has vastly different effects on cardiac function, many of which are not completely understood. The metabolic changes that occur during the progression of both obesity and diabetes can be monitored and evaluated in cardiomyocytes to investigate which cellular components are contributing to the sustainability of cells in these pathologic conditions.

Given its high relevance in cancer chemotherapeutics, it is of note to mention doxorubicin’s (DOX) profound cardiotoxicity effects, particularly regarding its effects on autophagy [92,93,94,95]. It has been proven efficacious against a wide variety of neoplasms. Despite its wide use, DOX has toxicity-limiting effects that can lead to both acute and chronic cardiotoxicity [96,97,98]. It has widespread mitochondrial effects, particularly on cardiolipin, calcium dysregulation, electron transport chain, and redox cycling [95,99,100,101,102]. Although many of these characteristics and effects of DOX on the heart and mitochondria have been established, the use of genetically encoded biosensor technology has not been utilized in this field. Establishing the ATP dynamic movement within the cardiomyocyte after DOX treatment would be enlightening, as DOX has multiple targets in the mitochondria and its effects on ATP dynamics may not be as anticipated. This would provide information that could lead to potential supportive therapies given with DOX to decrease cardiotoxicity and prevent both acute and long-term side effects.

The hiPSC-CMs have been increasingly utilized as a cardiac model for research, drug testing and discovery. As cellular metabolism is an integral component to the cell’s overall function, evaluating their metabolic profile during standard maturation and metabolic experiments are crucial in furthering their translational application and potential. There have been studies showing progressive remodeling of pathways in hiPSC-CMs during maturation, such as their increased ability to utilize fatty acids as a source of energy [103,104]. Current methodologies to test rates of glycolysis and OXPHOS include looking at aspects of metabolism such as lactate production [22,104,105], which is indicative of overall glycolysis. However, one of the downsides to these methods are that they are not direct, as the lactate is a byproduct of glycolysis. With an ATP biosensor, the real-time glycolytic activity can be determined in the cardiomyocyte or pluripotent precursor, alongside the evaluation of compartmental ATP content. Additionally, this ATP qualitative analysis could be performed over the maturation period of the cardiomyocyte to evaluate energy usage at various time periods.

Beyond cardiovascular disease, metabolic derangement is also a common feature in disease such as neurodegenerative disease and cancer. Devastating human neuromuscular disorders such as Parkinson’s and Alzheimer’s disease have been linked to defects in the ATP synthase and mitochondrial function [106,107,108,109]. In short, Parkinson’s disease (PD) is the degeneration of dopaminergic neurons within the substantial nigra pars compacta, resulting in movement-related symptoms, such as tremors, rigidity, slowness of movement, and poor balance [108,109,110,111]. Interestingly, PD is associated with defects in mitochondrial respiration, specifically at the level of complex I [112]. It was discovered that exposure to 1-methyl-4-phenyl-1,2,3,4-tetrahydropyridine (MPTP) selectively kills dopaminergic neurons, and results in an acute and irreversible parkinsonian syndrome [111,113,114]. In previous experiments, complex I activity must be reduced by more than 50% to cause significant ATP depletion in non-synaptic brain mitochondria. However, complex I activity is only reduced by 25–30% in PD patients [112,115]. This argues against the theory that ATP depletion leads to PD-related dopaminergic neurodegeneration. Using genetically encoded biosensors, we can further investigate the role of OXPHOS complexes and ATP dynamic movement in PD, perhaps revealing how dopamine-deficient neurons can maintain ATP production despite a deficiency of complex I activity. Another neurodegenerative disease, Alzheimer’s Disease (AD), is the most common neurodegenerative disorder, particularly in the aging population [116]. The etiopathology behind AD remains largely unclear due to AD’s highly complex and multifactorial nature. To summarize, the primary pathogenesis behind AD is believed to be the presence of amyloid plaques, neurofibrillary tangles, synaptic dysfunction, and inflammation [117,118]. Amyloid plaques, most commonly amyloid β (Aβ), accumulate within the brain and cause cell death [116]. In addition, mitochondrial dysfunction is shown in AD neurons and other cellular populations in various models and organisms [117,119,120]. For example, it has been described that Aβ plaques can directly affect the functioning of complex V [121]. However, many other mechanisms may be involved in the dysregulation of the ATP synthase in AD, as the α subunit of the ATP synthase of AD patients is lipoxidized when exposed to oxidative stress [119,121,122]. This dysregulation and effect on components of the respiratory chain and ATP synthase in AD indicate that there is an underlying metabolic deficiency that may be contributing to the swift decline seen in AD patients. An ATP biosensor might be impactful in helping to reveal the intricacies and effects of AD proteins and aggregates and their effects on specific complexes. This technology has promise in revealing metabolic discrepancies in complex neurodegenerative diseases, such as PD and AD. In cancer, it is known that increased extracellular ATP levels are characteristic of the microenvironment surrounding tumor sites, as they must meet their intense metabolic demands to proliferate and survive [123,124,125,126]. Conversely, depletion of intracellular ATP levels promotes the necrosis pathway in cells over the apoptosis process [127,128,129]. One group showed that MCF7 breast cancer cells cultured in cell media containing glucose, glutamine, and pyruvate displayed ATP production rates 1.6-fold higher than cells cultured in limited substrates [130]. Interestingly, this increase occurred mainly through faster oxidative ATP production, with little to no increase in glycolytic ATP production. Designing experiments utilizing an ATP biosensor in cancer cell types would allow for a better understanding of their metabolic responses in real-time, and this can be an essential tool for the characterization of cell homeostasis and assessing ATP production in different stages of the diseases [125,126].

A reliable method to track the movement of and visualize ATP usage and production in cardiomyocytes, neurons, and cancer cells may reveal in-depth mechanisms when evaluating metabolic measurements. If we could characterize the movement of dysregulated ATP between compartments under various pathological conditions, we would gain a more holistic understanding of the pathology, alongside discovering better potential targets for treatments of these diseases.

## 3. Current Methodologies for Visualizing ATP in Living Cells

In the past, spatiotemporal dynamic evaluation of ATP within individual cells was limited due to the lack of efficient and affordable technology. Thus, a quantitative analysis of ATP levels in vitro and in vivo at the single-cell level would investigate ATP dynamics of biological activities in numerous types of tissues and cells. The fundamental ATP dynamics occurring throughout the cell are still unclear in many pathological states, and even normal cellular compartmental fluctuations still remain unclear. Utilizing a tool to evaluate the spatiotemporal dynamics of ATP would fix this long-term issue.

Recently, significant work has been done to develop further technology in which ATP is measured, ranging from harnessing the applicability of aptamers to older methods such as chromatography and mass spectrometry [131,132,133]. Other methods such as detecting ATP via UV−visible absorption [134], magnetic resonance spectroscopy [131,135], or nuclear magnetic resonance [136] have also been developed. Some utilize small molecules such as nitrogen, allowing for the high specificity of ATP detection [131,137]. Unfortunately, these methods cannot quantify ATP concentrations within living organisms. For these methods, ATP calculations must be performed from isolations or assays that require the harvesting and breakdown of cells or tissues. It is impossible to measure ATP levels of tissues with traditional methods while still maintaining the integrity of cellular, particularly mitochondrial features [131].

Recent development has shown improvement in ATP imaging technology by utilizing various starting materials, such as small organic indicators, nanoparticles, and fluorescent probes [138]. For example, Magnesium green (MgGr) is a magnesium-sensitive, small organic indicator of ATP that can be utilized to detect ATP hydrolysis [139]. This method has been employed to determine the ATP/ADP exchange rate through the mitochondrial AAC transporter in isolated rat cardiomyocytes [1,139]. Furthermore, MgGr has been shown to be useful for studying malignant cells, which thrive in hypoxic environments and contain mitochondria with significant alterations [10]. However, these methods have multiple disadvantages. The specificity of MgGr to ATP is imperfect because it also has a moderate affinity to calcium, leading to possibly inaccurate results. Moreover, MgGr signal depends on dye concentration, which makes experiments harder to replicate and leaves room for further error [139].

Measuring total ATP levels within cellular compartmental pools in real-time presents a newer and more innovative approach to qualitatively analyzing ATP. Although this method is not precisely quantitative, it can be helpful in determining changes in ATP concentrations in one region of a cell compared to another in a variety of disease states. As most ATP biosensors discussed in this review are based on ATP indication via fluorescence or bioluminescence, the exact concentration or number of ATP molecules within a cellular compartment is not able to be determined. The overall goal of these assays and imaging methodologies are to evaluate and visualize dynamic ATP trends, such as usage and depletion, between cells and their sub-compartments. However, semi-quantitative evaluation of the relative ATP signal is feasible. The currently developed technologies that utilize this approach to ATP quantification are mainly genetically encoded biosensors [140,141]. In conjunction with a fluorescent or bioluminescent protein, most of these biosensors harness the folding capabilities of the ϵ subunit of the bacterial ATP synthase subunit. The bacterial ATP synthase protein comprises a β-barrel domain located at the N terminus and an α-helical domain with two α-helices located at the C terminus. Upon ATP binding, the two α-helices interact and refine their conformational structure of the ϵ subunit, leading to fluorescent/bioluminescent illumination, indicating that ATP is present. Overall, this subunit adopts two different conformations: open (ATP-free) or closed (ATP-bound) [142,143,144]. The uses and applicability of this technology are limitless. We will further discuss each group of genetically encoded biosensors, along with their applications and limitations.

### 3.1. Förster Resonance Energy Transfer (FRET)

Named after Theodore Förster, Förster resonance energy transfer is a method of imaging in which energy transfers between two fluorophores [145,146,147]. This energy transfer only occurs if the two fluorophores are less than 10 nm apart and if the donor’s emission spectrum overlaps with the excitation of the acceptor fluorophore protein (Figure 2A). FRET is used to develop biosensors, as it can emit a specific fluorescence when bound to the molecule that initiates a conformational change. Specifically, biosensors have been developed to allow imaging when ATP is bound to the bacterial ATP synthase subunit [143,146,147,148,149]. The most common FRET ATP biosensor is ATEAM (Adenosine 5′-Triphosphate indicator based on ϵ subunit for Analytical Measurements), which comprises the ϵ subunit of bacterial ATP synthase wedged between a cyan and yellow donor and acceptor pair [146]. These fluorescent proteins are positioned at the N and C terminals, respectively. The binding of ATP triggers the conformation change that decreases the proximity between fluorophores, resulting in a higher spectral emission and illuminating as a new color. This method allows for the visualization and monitoring of intracellular ATP. FRET has been successfully applied in HELA cells [42], and human skin fibroblasts [150] and utilized to show the difference in ATP between cellular compartments [150]. More recently, ATP dynamics have been indicated in vivo by generating an engineered novel mouse line that expresses the FRET-based ATP biosensor in all tissues using a two-photon microscope [151].

The major concern when utilizing a FRET biosensor is that they are sensitive to pH [150]. The sensitivity of the donor/acceptor pair can lead to instability of the biosensor, leading to false interpretations of ATP levels. This should be considered especially in instances where acidic cellular compartments, such as lysosomes, are the main cellular component of interest. Other than common issues related to fluorescent imaging, such as autofluorescence and photobleaching, another shortcoming of the FRET biosensor is its relatively low signal-to-noise ratio due to detector noise and optical noise. These issues pose challenges in interpreting results from cell compartments with low ATP contents.

### 3.2. Bioluminescence and BRET

Bioluminescence is another biosensor technique that exploits the emitted light released from the enzymatic reaction. Whenever luciferase is in the presence of O_2_, ATP, and Mg^2+^, an oxidation reaction will occur that stimulates the release of photons. The emission spectra can vary depending on the type of luciferase used. However, it typically comprises spectra between 546 nm and 618 nm [152]. This technology harnesses the biology of bioluminescence to indicate the amount or concentration of ATP in plasma, tissue, or mitochondria. With targeting sequences added to the proteins, these indicators could localize to specific cellular compartments, such as the mitochondria or endoplasmic reticulum. Numerous variations of bioluminescent techniques have been developed, some for more specific applications. One group developed a luciferase-enzymatic reaction system called Syn-ATP (Figure 2D) composed of synaptophysin, a synaptic vesicular protein, fused to mCherry and luciferase [153]. mCherry provides a method to determine the luminescence/fluorescence ratio. As more ATP is consumed, the bioluminescence becomes more intense.

BRET, or bioluminescence resonance energy transfer, is similar to FRET but is instead based on the energy transfer between a bioluminescent molecule and a fluorophore. When the bioluminescent protein and fluorophore are close, less than 10 nm apart, the luciferase enzymatic reaction excites the fluorophore, and spectra are emitted [154,155]. One of the first indicators using this method was developed utilizing the ATP binding domain of the ϵ subunit of bacterial ATP synthase that is wedged between a yellow fluorescent protein called Venus (emission 528 nm) and NanoLuciferase (Figure 2B). A unique advantage to NanoLuciferase is that it does not require ATP as a cofactor to function—it simply binds to ATP, inducing a conformational change in the biosensor. After ATP binding, NanoLuciferase converts furimazine to furimamide. This particular subset of luciferase removes the potential for inaccurate interpretations of ATP levels in the cell due to inadvertent consumption. Another similar method was developed in which the luciferase enzyme is split into two portions, and ATP binding thus rejoins the subunits, leading to a BRET reaction [156].

As some forms of luciferase require ATP as a cofactor to function, it is important to discern which derivative of luciferase would best suit an experiment, as this could lead to misinterpretation of ATP amounts within the cell. One of the significant issues associated with bioluminescent-utilizing biosensors is that some luciferases also have an affinity for ADP. The lack of specificity to only ATP leaves room for error in determining the amount of ATP produced [157]. Furthermore, certain drugs and other cellular metabolites can impact luciferase enzymatic activity, resulting in potentially unreliable and inconsistent results [158]. Lastly, as the enzyme activity of luciferase is oxygen-dependent, this method cannot be utilized in hypoxic conditions [68,158].

### 3.3. Single, Ratiometric, and Intensiometric Biosensors

Another method in developing biosensors is to utilize a single fluorophore instead of two. Many biosensors are developed based on ratiometric and intensiometric properties, indicating that they either exhibit a single fluorescent signal or lead to an increase in fluorescence emission at their corresponding wavelengths. The difference in fluorescence intensity can then be analyzed to determine ATP content in cells and subcellular compartments qualitatively. In these cases, the increase in fluorescence only occurs when a particular molecule, such as ATP, is bound, thus changing the conformation. Small changes in the circularly permutated fluorophore occur at the N and C terminal of the protein which induce a change in conformation, altering the emission properties of the fluorophore [127].

One particular biosensor, QUEEN, is a genetically encoded, ratiometric fluorescent ATP indicator that has been utilized to determine the global ATP levels within bacteria and yeast [159]. QUEEN, or Quantitative Evaluator of cellular ENergy, is composed of a single, circularly permutated enhanced GFP wedged between two α-helices of the ϵ subunit of bacterial ATPase (*Bacillus subtilis*). When ATP is bound, the biosensor slightly changes its confirmation, and a shift of the 400/494 nm ratio is seen proportional to the amount of ATP present in the cell. Another ATP biosensor with a single fluorophore is iATPSnFR (Figure 2C), an intensity-based ATP-sensing fluorescent reporter which monitors extracellular and cytosolic ATP [160]. This indicator has the same components as in QUEEN. However, unlike QUEEN, iATPSnFR is more sensitive to pH, which poses potential problems and limitations for its use in specific cell compartments with large fluctuations in pH.

There have been advances in the utilization and optimization of ATP biosensors to target specific compartments of the cell. One particular group designed a family of biosensors called MaLions (Figure 2E), or Monitoring aTP Level intensiometric turn-on indicators [161]. These biosensors consist of the ϵ subunit of bacterial ATP synthase, with red, green, and blue fluorophores. These were adapted to target and follow the cytosolic, mitochondrial, and nuclear ATP levels, respectively. This family of biosensors is beneficial, as their organelle-specificity to ATP allows for evaluating ATP dynamics and flux between compartments. Additionally, this family is intensiometric, meaning that the higher the fluorescent intensity, the higher the ATP concentration. This group of biosensors also has a low sensitivity to pH. MaLions provide an exciting and innovative future for evaluating ATP in vitro and in vivo.

Perceval (Figure 2F) is a family of ratiometric biosensors using the ADP/ATP ratio to quantify ATP in living cells [162]. Perceval’s fluorescence response is related to the ratio of ATP to ADP levels [163]. Although the absolute individual amounts of ATP, ADP, and AMP might vary widely within the cells, the ratios of [ATP]/[ADP] and [ATP]/[AMP] might be considered a more reliable indicator of metabolic activity between compartments and from cell to cell [164,165]. This biosensor family is composed of a bacterial regulatory protein, GlnK1, linked to a circularly permuted Venus fluorescent protein. The structure of GlnK1 is similar to the ϵ subunit of bacterial ATP synthase. It undergoes a conformational change whenever ATP is bound, resulting in a change of the Venus protein’s excitation/emission of the 490/405 nm ratio. However, ADP does not induce a conformational change [162,163,164,166].

Compared to other ATP indicators, genetically encoded biosensors provide a unique approach to real-time imaging. They allow for the discovery and determination of living and single-cell spatiotemporal ATP dynamic evaluation, providing a long-term approach to evaluating specific pathologies in cell culture. Moreover, biosensors with circularly permuted fluorophores allow for detecting small changes in specific compartments of a single living cell. Perhaps even more so, this technology is relatively affordable, and experiments can be conducted with standard microscopy technologies. Genetically encoded biosensors provide broad usability in various biological fields of research. The ease of use, affordability, and lack of needing super high-tech microscopy provides a more accessible way to conduct better and more innovative research. With various types of ATP biosensors for use in living cells and in vivo experiments, it is essential to determine the one that would best suit a particular study (Table 1). Although a significant amount of work has been carried out to develop and optimize these methods, their applicability has not been utilized to their full potential to answer important biological questions.

**Table 1 cells-11-01920-t001:** Comparison of unique features seen with genetically encoded ATP biosensors.

Genetically Encoded ATP Biosensors				
Biosensor	Technique	Mechanism	Advantage	Disadvantage
**ATEAM** [146]	**FRET**	Adenosine 5′-triphosphate indication based on ϵ subunit for analytical measurement; ATP binding causes an increase in Forster resonance energy transfer between a CFP and YFP and results in a higher wavelength release; comprised of bacterial ϵ subunit of bacterial ATP synthase with cyan and yellow donor/acceptor pairs at N and C terminals, respectively	Qualitative/quantitative; spatiotemporal resolution	Sensitive to acidic pH, thus limiting which cellular subcomponent cell can use; can undergo glycosylation in ER and Golgi which inhibits its ability to bind to ATP
**GO-ATEAM** [143]	**FRET**	Similar to ATEAM but CFP and YFP are replaced by green (GFP) and orange (mKOk) fluorescent pair, respectively		
**BTEAM** [155]	**BRET**	Composed of e subunit of bacterial ATP synthase flanked by Venus at the N terminal and Nanoluciferase at the C terminal; emitted light is produced by Nanoluciferase because oxidation of luciferin cases emission of photons; capacity of luciferin to emit light is directly correlated to amount of ATP available	Qualitative/quantitative; spatiotemporal resolution; no need for laser, as light emission come from enzymatic reaction after administration of luciferase substrate; avoid generation of autofluorescent and phototoxicity; very sensitive; simplicity of assay; can add localization signals to target cell subcompartments	Luciferin limitation due to inhibition of reaction from other drugs; limits potential with some drug development; enzymatic and substrate concentration limitations; transfection efficiency limitations; optimization required for maximal detection; some luciferases produce ATP from pools of ADP
**ARSeNL** [167]	**BRET**	ATP detection via ratiometric mScarlet-NanoLuc sensor, similar to BTEAM		
**QUEEN** [168]	**Ratiometric**	Quantitative evaluator of cellular energy; cpFP is inserted between two a helices of ϵ subunit of ATP synthase with linkers	similar results to bioluminescence luciferase assays	Modest pH sensitivity
**iATPSnFR** [160]	**Intensiometric**	Intensity-based ATP-sensing fluorescent reporter consists of circularly superfolder GFP between 2 alpha helices of ϵ subunit of bacterial ATP synthase; when ATP binds, rapid increase in fluorescence occurs	spatiotemporal resolution	Modest pH sensitivity
**Syn-ATP** [153]	**Bioluminescence**	Luciferin-reaction based; synaptophysin targets synaptic vesicle proteins and mCherry helps to determine total amount of luciferase using a luminescence/fluorescent ratio	Qualitative/quantitative; only used for synaptic vesicles	No spatiotemporal resolution; some luciferases produce ATP from pools of ADP
**MaLion** [161]	**Intensiometric**	multiple constructs created to target subcellular compartments (cytosol, mitochondria, nucleus); consists of a fused ϵ subunit of bacterial ATP synthase to red, blue, or green	Qualitative/quantitative; spatiotemporal resolution; has organelle-targeted specific ATP estimations; the higher the ATP, the brighter the fluorescence; low pH sensitivity	Potential phototoxicity due to fluorescence emission in living cells; transfection efficiency in hard to transfect cells
**Perceval** [164]	**Ratiometric**	Based on estimation of ADP/ATP; composed of GlnK1 (a bacterial regulatory protein) linked to Venus; GlnK1 undergoes a conformational change when bound to ATP	Qualitative/quantitative; spatiotemporal resolution; no conformational change when bound to ADP	Some pH sensitivity

## 4. Perspective

The main goal for developing and utilizing innovative technology, such as genetically encoded ATP biosensors, is to gain further mechanistic insights into cellular energy metabolism in health and disease. Genetically encoded biosensors are relatively inexpensive, requiring only the plasmid, a transfection or transduction method, the cell line of interest, and machinery to capture and analyze the fluorescence or bioluminescence.

The current cost of using an animal model, even mice, is exponentially higher than first testing the hypothesis in cell culture. A cell-based model of disease has the potential to be used to screen for potential drugs and their effects on cellular metabolism, even before moving the preliminary studies into an animal model. Using an ATP biosensor to evaluate the effects of drugs would reduce overall costs and eliminate inactive compounds before progressing further with testing.

Based on current limitations and inevitable troubleshooting issues with fluorescent and bioluminescent biosensors, a few necessary features would eliminate many issues and help make the “perfect” biosensor. First, a range of sensors with varying binding affinities to ATP would allow for more specific, qualitative measurements within the cellular ATP pools across various compartments and organelles. Next, faster binding of ATP to the biosensor would allow for faster and more accurate illumination of the indicator, thus providing a more precise, real-time response to a particular drug or metabolic effector. Another essential feature of an effective biosensor would be higher brightness and contrast ratios. This would be another method to provide more accurate changes in ATP fluctuations and spatiotemporal dynamic changes. An additional approach to measuring ATP would be the indirect measurement of ATP hydrolysis products. This may be beneficial in cases where the rate and amount of ATP usage are more important to evaluate than total ATP amounts. Lastly, most genetically encoded biosensors discussed in this review only have a few colors, such as cyan, GFP, and mCherry. It would be beneficial to have a broader spectral color variation, allowing for multiple compartments to be evaluated simultaneously. With dual-compartmental ATP compartmental and organelle visualization, the change between compartments after metabolic effectors can be evaluated, leading to a direct analysis of understanding which cellular components are responsible for compensation in different physiologic or pathologic scenarios. This method of evaluating and analyzing ATP regulation bears unlimited possibilities for cellular metabolic research. It allows us to answer questions in real-time that have not been thoroughly answered yet.

One of the main drawbacks of real-time imaging analysis is low throughput. Deep learning imaging analysis has been flourishing in recent years. The combination of this new development may help to partially ease the analysis workload and improve objectivity and reproducibility. Furthermore, simultaneous or parallel analyses of ATP signals with other crucial biological parameters, such as calcium, pH, and ROS, should open new windows to previously unknown biological interactions during energy metabolism with different compartments of cells. For example, Wu et al. recently developed a dual-channel fluorescent probe for the simultaneous monitoring of peroxynitrite and ATP [169]. These innovative technologies enable spatiotemporal analyses of ATP dynamics along with other key metabolites in living cells under physiological and pathological contexts.

In summary, we have discussed the overall regulation and transport of ATP under normal conditions and general pathologic conditions. Then, we briefly covered the evolution of technology and protocols for ATP detection and quantification, focusing specifically on genetically encoded ATP sensors. We also discussed potential ways that ATP biosensors can be used in potential therapeutic studies in cell culture to validate the direct cellular effects of the treatments before progressing to more expensive and time-intensive animal models.

## Figures and Tables

**Figure 1 cells-11-01920-f001:**
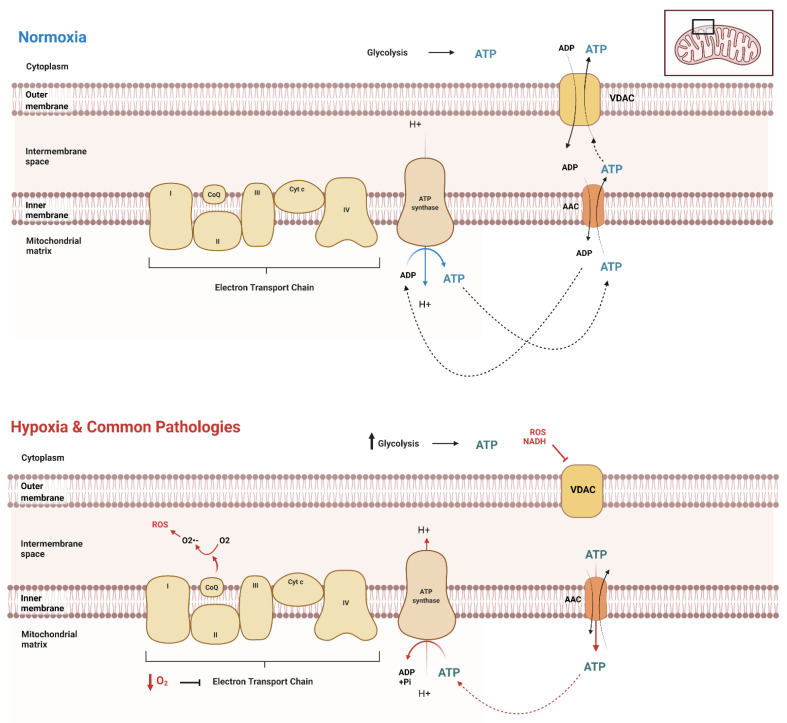
ATP movement between mitochondria and cytosol. Under normal conditions (normoxia), the cell has distinct cellular processes in which to transport ATP throughout the cell, both into and out of the mitochondrial matrix. The mitochondria produce the majority of ATP in normoxic conditions, while glycolysis produces a significantly lower amount. However, in the setting of hypoxia or similarly acting pathologies, there is an upregulation of glycolysis, as the decrease in oxygen causes a significant downregulation of electron transport chain function. This results in the reversal of AAC transport of ATP to inside the matrix, where it is then hydrolyzed by the also reversed rotation of ATP synthase.

**Figure 2 cells-11-01920-f002:**
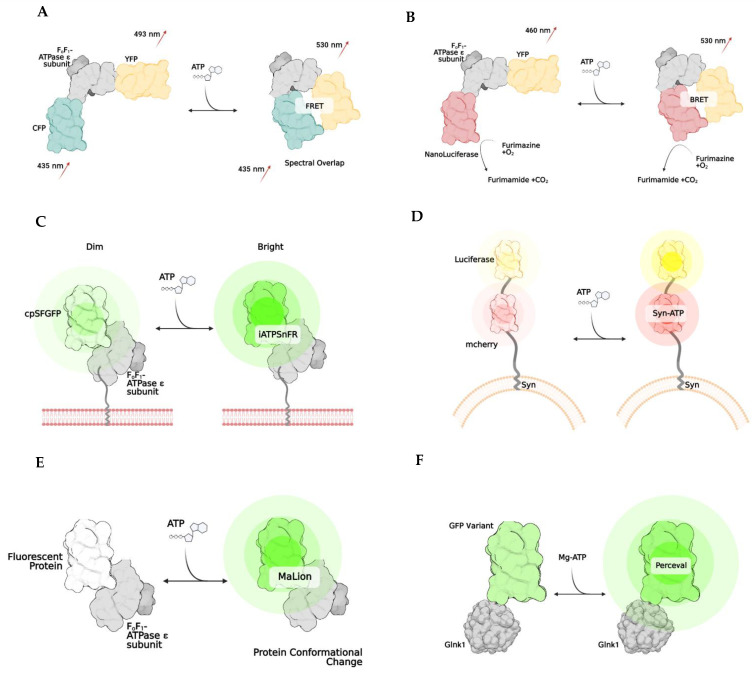
Examples of genetically encoded biosensors. Förster resonance energy transfer, or FRET, (**A**) works by changing emission spectra after the binding of a substance (**A**). The result is due to the two fluorophores being brought closer together (<10 nm). Bioluminescent resonance energy transfer, BRET, works similarly to FRET, except NanoLuciferase becomes active once the substrate of interest is bound, resulting in a change of emission. (**B**) iATPSnFR is a unique biosensor, in which the ATPase subunit has a membrane-bound region, which allows it to be anchored to a cell’s surface. (**C**) Syn-ATP was designed to target synaptic nerve endings, as the synaptophysin protein anchors the biosensor to a synaptic vesicle membrane. Once ATP is bound, the mCherry and Luciferase enzymes provide a fluorescent/luminescent ratio for analysis. (**D**) MaLions are intensiometric biosensors, fluorescing brighter as more ATP is bound. These are unique as multiple biosenors have been designed in this family to target specific cellular organelles, making the ATP levels visible across multiple compartments. (**E**) Lastly, perceval is a ratiometic indicator that indicates its ATP level based on the ratio of ADP to ATP (**F**).

## Data Availability

Not applicable.

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
