# Peer review of "Genetically Encoded ATP Biosensors for Direct Monitoring of Cellular ATP Dynamics"

_cells, 2022, doi:10.3390/cells11121920_

Round 1

Reviewer 1 Report

This is a good review that nicely summarizes the mechanism and its significance of ATP in cytoplasm and mitochondria. However, ATP concentration is an important issue not only for cytoplasm and mitochondria, but also for ATP accumulation in secretory granules in purinergic chemical transmission. Thus, I think it's better to briefly touch on these issues.

Author Response

DW

Reviewer 2 Report

This review by White and Yang describes some of the current methodologies for visualizing ATP in living cells. As this is an emerging filed, this review is useful and fills a void. This reviewer has some suggestions that may help improve the utility of this manuscript.

1.     Line 95-102. The authors state that major transporters in mitochondria are the ANT and the ATP-Mg/Pi carrier, then later in the paragraph mention VDAC. Reconcile the description of 3 transporters but only 2 are presented in Figure 1.

2.     Some additional attention should be given to quantification. For instance, the authors state that “this method is not precisely quantitative” (line 339-340), but is in not clear what this means. Is it possible to quantify a mM amount of ATP using biosensors? If so, how might this be done. Alternatively, are these methods largely limited to dynamics and relative changes between cell or cellular compartments? Some specificity on this issue may help the reader to make a better choice on the method of ATP measurement.

3.     Bioluminescence and BRET (section 3.2) seem unique among the biosensors since they consume ATP (versus just binding ATP for the others). Could the authors comment on whether this consumption might be expected to change the balance of ATP consumption versus ATP production? What effect does consuming ATP have on the measure of ATP?

4. It would be useful for the authors to put references in the Table that directly guide the reader to the important manuscripts of each biosensor.

Author Response

DW
